# Preoperative Typing of Thyroid and Parathyroid Tumors with a Combined Molecular Classifier

**DOI:** 10.3390/cancers13020237

**Published:** 2021-01-11

**Authors:** Sergei E. Titov, Evgeniya S. Kozorezova, Pavel S. Demenkov, Yulia A. Veryaskina, Irina V. Kuznetsova, Sergey L. Vorobyev, Roman A. Chernikov, Ilya V. Sleptsov, Nataliya I. Timofeeva, Mikhail K. Ivanov

**Affiliations:** 1Department of the Structure and Function of Chromosomes, Institute of Molecular and Cellular Biology, SB RAS, 630090 Novosibirsk, Russia; microrna@inbox.ru (Y.A.V.); ivanovmk@vector-best.ru (M.K.I.); 2AO Vector-Best, 630117 Novosibirsk, Russia; 3Department of Natural Sciences, Novosibirsk State University, 630090 Novosibirsk, Russia; demps@math.nsc.ru; 4National Center of Clinical Morphological Diagnostics, 192283 Saint Petersburg, Russia; kozorezovaes@yandex.ru (E.S.K.); k.iriniya.v@mail.ru (I.V.K.); slvorob@gmail.com (S.L.V.); yaddd@yandex.ru (R.A.C.); natalytim@mail.ru (N.I.T.); 5Institute of Molecular Pathology and Pathomorphology, 630117 Novosibirsk, Russia; 6Institute of Cytology and Genetics, SB RAS, 630090 Novosibirsk, Russia; 7Department of Endocrinology and Endocrine Surgery of Saint Petersburg State University N.I. Pirogov Clinic of High Medical Technologies, 190103 Saint Petersburg, Russia; newsurgery@yandex.ru

**Keywords:** thyroid carcinoma, parathyroid adenoma, preoperative diagnosis, molecular marker, HMGA2, microRNA, mitochondrial DNA

## Abstract

**Simple Summary:**

We previously proposed a new diagnostic algorithm that allows identification and classification of malignancy markers of thyroid tumors in cytological preparations of biopsy material through an analysis of several molecular markers. We previously evaluated the diagnostic characteristics of this algorithm on a sample of category III and IV cytological preparations (Bethesda system, 2017) for the detection of malignant tumors. However, in that study, we did not determine the accuracy of classification. Also, the algorithm did not allow discrimination of parathyroid gland nodules. In the present work, our goal was to include the identification of parathyroid cells in the molecular classifier and to evaluate the performance of our algorithm on the typing of thyroid tumors. We demonstrated that the diagnostic panel including the analysis of microRNA and mRNA expression, the V600E mutation in the BRAF gene, and mitochondrial-to-nuclear DNA ratio enables accurate identification of parathyroid and several types of thyroid carcinomas.

**Abstract:**

In previous studies, we described a method for detecting and typing malignant tumors of the thyroid gland in fine-needle aspiration biopsy samples via analysis of a molecular marker panel (normalized *HMGA2* mRNA level; normalized microRNA-146b, -221, and -375 levels; mitochondrial-to-nuclear DNA ratio; and *BRAF*^V600E^ mutation) in cytological preparations by quantitative PCR. In the present study, we aimed to estimate the specificity of the typing of different thyroid tumors by the proposed method. Fine-needle aspiration cytological preparations from 278 patients were used. The histological diagnosis was known for each sample. The positive and negative predictive values of the method assessed in this study were, respectively, 100% and 98% for papillary thyroid carcinoma (*n* = 63), 100% and 100% for medullary thyroid carcinoma (*n* = 19), 43.5% and 98% for follicular carcinoma (*n* = 15), and 86% and 100% for Hürthle cell carcinoma (*n* = 6). Thus, we demonstrate that the diagnostic panel, including the analysis of microRNA expression, mRNA expression, the *BRAF*^V600E^ mutation, and the mitochondrial-to-nuclear DNA ratio, allows the highly accurate identification of papillary thyroid carcinoma, medullary thyroid carcinoma, and Hürthle cell carcinoma but not malignant follicular tumors (positive predictive value was below 50%).

## 1. Introduction

In terms of prevalence, nodules of the thyroid gland are predominant among pathologies of the endocrine system: they occur in 5–8% of the population, and when ultrasonography is applied, this proportion increases to 15–67% [1,2]. These lesions range from hyperplastic or adenomatous tumor-like masses (including nodular goiter and nodular transformation in chronic autoimmune thyroiditis) to tumors (including encapsulated ones such as benign follicular thyroid adenoma (FTA) and Hürthle cell adenoma (HCA)) and malignant tumors, including well-differentiated cancers (follicular thyroid carcinoma (FTC), Hürthle cell carcinoma (HCC), and papillary thyroid carcinoma (PTC)), medullary thyroid carcinoma (MTC), poorly differentiated thyroid carcinoma (PDTC), and anaplastic (undifferentiated) thyroid carcinoma (ATC).

Diagnostic evaluation of thyroid nodules includes ultrasonographic examination of the thyroid gland and lymph nodes of the neck. Ultrasonography permits the detection of nodules less than 1 cm, which may not be detected by palpation; in such cases, in accordance with the current guidelines, it is necessary to perform an ultrasound-guided biopsy followed by cytological examination of the aspirate [3,4,5].

The Bethesda System for Reporting Thyroid Cytopathology is most commonly used to make cytological diagnoses and involves six diagnostic categories stratified by malignancy risk [6]. Although the cytological assessment is fairly accurate in many cases, approximately 15–20% of the aspirates fall into the indeterminate diagnostic category (Bethesda III, IV, or V) [7]. In this heterogeneous group, for objective reasons, it is impossible to accurately determine the degree of malignancy of thyroid tumor nodules on the basis of cytomorphological characteristics alone. Thus, the nodules belonging to Bethesda categories III and IV (2017) pose a known clinical problem [8]. According to clinical recommendations, most patients with an indeterminate cytological diagnosis (including all those belonging to the Bethesda IV category) are referred for lobectomy or molecular testing [6]. Notably, approximately 70–80% of operated thyroid nodules are benign according to the results of postoperative histological examination [3,9].

In recent years, molecular testing has been increasingly used to improve diagnosis and optimize the treatment of patients with thyroid nodules who have an indeterminate cytological diagnosis. Meanwhile, molecular testing has transitioned from small panels of mutations to next-generation sequencing, multigene classifiers, and the use of other molecular markers [10,11]. Many authors think that it is important for modern molecular tests to accurately distinguish thyroid nodular lesions, medullary carcinomas, and parathyroid lesions [12].

The main molecular tests employed in the diagnosis of thyroid lesions conform to these trends. For example, the Afirma Genomic Sequencing Classifier (GSC) includes eight components: modules of the parathyroid gland and of medullary cancer, *BRAF*^V600E^ mutation detection (PTC), detection of *RET–PTC1* and *RET–PTC3* translocations (PTC), a follicular content index (identifies samples with a low thyroid gland cell count), an ensemble model (a suspected cancer/benign sample), a Hürthle cell index, and the Hürthle cell neoplasm index [13]. Another molecular test, ThyroSeq v3, consists of the following: detection of parathyroid cells, C cells (MTC), and nonthyroid cells, and a genomic classifier (negative/positive) [14].

We previously proposed a new diagnostic algorithm that allows identification and classification of malignancy markers of thyroid tumors in cytological preparations of biopsy material through an analysis of the following molecular panel: the *HMGA2* oncogene; microRNA (miRNA, miR) 146b, miR-221, and miR-375; the ratio of mitochondrial DNA (mtDNA) to nuclear DNA (nDNA); and *BRAF*^V600E^ mutation detection [15,16]. The proposed algorithm was not originally designed for the typing of thyroid neoplasms, but because it was created to detect cancer (rule-in test), specific markers were selected for each type of malignant tumor in those studies. This approach has ultimately enabled us to use this algorithm as an additional tool for the classification and diagnosis of tumors. For instance, the *BRAF*^V600E^ mutation and increased expression of miR-146b have been found to be specific markers of PTC. MTC has proved to be characterized by a significantly increased level of miR-375 expression. For example, in PTC, the expression level of miR-375, on average, is 30- to 40-fold higher compared to nontumorous lesions; in MTC, it is higher than that in PTC by two orders of magnitude, on average [15]. Only for follicular carcinoma have no specific markers been found; in this algorithm, it is identified by exclusion using nonspecific markers of malignancy: increased expression of the *HMGA2* oncogene in combination with miR-375 or miR-221. Increased levels of miR-221 and mtDNA were found to be a marker of HCC [15].

We used the ratio of mtDNA to nDNA because Hürthle cells (oncocytic cells) feature the accumulation of a large number of abnormal mitochondria regardless of the organ of origin or the benign or malignant nature of the lesion [17,18]. Hürthle cells can be observed in Hashimoto’s thyroiditis, nodular goiter, HCA, HCC, PTC, and PDTC; only in ATC are they extremely rare [19]. Hürthle cell (oncocytic, oxyphilic) subtypes of PTC and FTC have molecular features and prognoses that are similar to those of their traditional counterparts [20,21]; however, the oncocytic subtypes show a lower propensity to accumulate iodine and are therefore less sensitive to radioiodine therapy. According to the WHO (2017), HCA and HCC are categorized as independent oncological entities that are distinct from thyroid follicular tumors [22]. The use of the mtDNA/nDNA ratio in our test not only allows detection of HCC but also helps with the verification of oncocytic subtypes of other tumors (PTC and MTC), HCA, and benign nodules containing Hürthle cells (Hashimoto’s thyroiditis).

The previously described version of the classifier [16] does not discriminate thyroid and parathyroid tumors. During an examination of samples of the parathyroid gland (adenoma), they were usually identified as follicular thyroid neoplasms with markers of malignancy (FN-MM). The FN-MM concept is described in our previous work [15]; in brief, being unable to find a marker distinguishing some follicular variant of PTC (FV-PTC) cases from FTC, we combined them into one group. Thus, even though parathyroid tumors are relatively rarely biopsied on purpose, it was necessary to add a marker to the classifier to distinguish between thyroid and parathyroid cells. According to the analysis of published data, the *GCM2* gene (the human ortholog of the *Drosophila* glial cells’ missing gene) was chosen as such a marker, which is predominantly, if not exclusively, expressed in proliferating [23] and mature parathyroid cells [24].

We previously evaluated the diagnostic characteristics of the above algorithm on a sample of category III and IV cytological preparations (Bethesda system, 2017) for the detection of malignant tumors [16]. Nonetheless, in that study, we did not determine the accuracy of classification. In addition, the algorithm did not allow discrimination of parathyroid gland nodules. In the present work, our goal was to include the identification of parathyroid cells in the molecular classifier and to evaluate the performance of our algorithm on the typing of thyroid tumors. Among specimens of the Bethesda III–IV categories, it is difficult to find sufficient numbers of all the tumor subtypes that we needed for this study; therefore, in this analysis, we included clinical specimens from all Bethesda categories. Accordingly, in the present study, we validated the previously described typing algorithm [15] using an independent study population completely unrelated to the one that was used in our previous studies [15,16].

## 2. Results

### 2.1. Adding a Parathyroid Marker to the Classifier

Our results confirmed that *GCM2* mRNA is detectable in parathyroid samples and in a very small number of thyroid samples (Figure 1). Using a receiver-operating characteristic analysis, a cutoff was calculated (0.168) at which parathyroid and thyroid tumors are discriminated with 100% sensitivity and specificity.

### 2.2. Increased Expression of the HMGA2 Gene Is a Marker of Malignant Tumors Originating from Follicular Cells

The expression levels of *HMGA2* in different types of tumors are shown in Figure 2.

The results we obtained in this work are fully consistent with those of our previous study [15]: an increase in *HMGA2* expression (compared to goiters) is characteristic of PTC (*p* = 3.9 × 10^−14^), FTC (*p* = 4.98 × 10^−6^), ATC (*p* = 1.72 × 10^−5^), and, to a lesser extent, noninvasive follicular thyroid neoplasm with papillary-like nuclear features (NIFTP, *p* = 0.00015), but not typical for MTC (*p* = 0.702) and parathyroid adenoma (PTA, *p* = 0.0681). Among the FTAs, however, there were several (*n* = 8; 7.2%) samples with *HMGA2* expression exceeding the cutoff used in our classifier. In anaplastic cancer, *HMGA2* expression was also found to be elevated in most cases (80%, *p* = 0.000017).

### 2.3. Representative Levels of miRNA Expression Differ among the Analyzed Types of Tumors

The relative expression levels of three miRNAs in different types of tumors and lesions are illustrated in Figure 3.

The level of miR-146b was found to be significantly higher (compared to goiters) in PTC (*p* = 1.2 × 10^−14^) and, in decreasing order, in ATC (*p* = 0.000093) and NIFTP (*p* = 0.0279). The level of miR-221 is increased in many thyroid tumors (ATC, *p* = 0.000257; MTC, *p* = 3.22 × 10^−7^; PTC, *p* = 1.03 × 10^−12^; and FTC, *p* = 0.0146), and especially in HCC (*p* = 0.000121), where it was elevated in all specimens. In medullary cancer (compared to all other tumors), the expression of miR-375 is significantly increased (PTC, *p* = 6.19 × 10^−11^; ATC, *p* = 1.45 × 10^−5^; NIFTP, *p* = 1.45 × 10^−5^; HCC, *p* = 3.24 × 10^−4^; and FTC, *p* = 8.43 × 10^−8^), on average, by two orders of magnitude. If we consider all the tumors without MTC, then the level of miR-375 (compared to goiters) is higher in PTC (*p* = 4.23 × 10^−12^), ATC (*p* = 0.004), and parathyroid tumors (*p* = 1.32 × 10^−4^), which is why they were identified as malignant tumors by the previous version of the classifier.

The miRNA levels data are shown in Appendix A separately for different PTC subtypes and for NIFTP. The expression of all three miRNAs can be ranked as follows (in decreasing order): tall-cell variant of PTC (TCV-PTC), classic variant of PTC (CV-PTC), FV-PTC, solid variant of PTC (SV-PTC), and NIFTP, with one deviation from this pattern: the level of miR-146b was higher, on average, in CV-PTC than in TCV-PTC (Appendix A). Notably, according to the profile of these miRNAs, NIFTP is closer to SV-PTC than to FV-PTC (Appendix A).

### 2.4. Identification of the V600E Somatic Mutation in the BRAF Gene

The V600E mutation of the *BRAF* gene was detectable only in papillary cancers (58.7%; *n* = 63) and anaplastic cancers (40%; *n* = 10). The prevalence of this mutation in different types of PTC, NIFTP, and ATC is detailed in Appendix A. Thus, according to our data, in TCV-PTC, the *BRAF*^V600E^ mutation was detectable in 100% of cases (*n* = 8), in 79.2% of cases in CV-PTC (*n* = 24), in 40.9% of cases in FV-PTC (*n* = 22), and in 11.1% of cases in SV-PTC (*n* = 9); in samples of NIFTP (n = 10), the *BRAF*^V600E^ mutation was not detectable. Just as in the miRNA analysis, the percentage of samples with the *BRAF*^V600E^ mutation could be ranked in decreasing order as follows: TCV-PTC, CV-PTC, FV-PTC, SV-PTC, and NIFTP, with complete disappearance in the latter.

### 2.5. The Ratio of mtDNA to nDNA Is Significantly Higher in HCCs

As expected, the mtDNA/nDNA ratio was markedly increased in HCCs (Figure 4).

Some samples with a large amount of mtDNA (above the cutoff set by us) were identified among goiters, PTCs, and especially FTAs. It was noted in the histological report that nine cases of FTA were oncocytic; among them, in eight cases (88.9%), the mtDNA/nDNA value exceeded the cutoff specified in our classifier. In five other samples (4.9%) in which mtDNA/nDNA exceeded the cutoff (one of them was identified as HCC by the classifier), histological examination showed a usual FTA subtype. In ATC, the amount of mtDNA was the lowest.

### 2.6. Typing of Thyroid Tumors by the Molecular Classifier

The decision tree used in this study is depicted in Figure 5.

We analyzed 278 samples using the molecular classifier, of which 10 were parathyroid tumors, 10 ATC, and 10 NIFTP according to histological analysis. Although the latter two groups were analyzed here, these data were not included in the calculation of the classifier’s diagnostic characteristics because it was not trained to classify them. The flow of clinical samples through this study, in accordance with the Standards for Reporting of Diagnostic Accuracy Studies, is shown in Appendix A.

The results of identification of PTA samples using the new marker (*GCM2*) are described above. In our study population, the accuracy of detecting parathyroid tumors was 100%.

The stratification of thyroid gland samples by the presented method showed that the following pathologies were identified correctly: 19 out of 19 MTC samples (100%); 59 of 63 PTC samples (93.6%), with all discrepancies attributable to SV-PTC; 6 of 6 HCC samples (100%); 10 of 15 FTC samples (66.7%); 33 of 34 goiter samples (97%); and 100 of 111 FTA samples (90.1%; Table 1).

The achieved diagnostic characteristics for detecting cancer and different types of thyroid malignant tumors are detailed in Table 2.

A complete distribution of the decisions of the molecular classifier on different lesions and tumors of the thyroid gland, including ATC and NIFTP, can be found in Appendix A.

Of the 10 ATC samples, 8 were assigned by the molecular classifier to PTC and 2 to FN-MM. Of the 10 NIFTP samples, 3 samples were identified as PTC, 1 as FN-MM, and 6 as the benign nodule group.

## 3. Discussion

The previously described version of the classifier [15,16] was trained on a dataset that did not contain parathyroid tumors. As revealed by subsequent testing of the classifier, it identifies these tumors as FN-MM. This method is supposed to be applied to clinical samples on which cytological analysis has been completed. However, discriminating the lesions of the parathyroid gland from those of the thyroid gland can be challenging due to the similarity in morphological features and anatomical proximity. Up to 0.4% [25] of the nodules that are classified by biopsy as thyroid nodules actually belong to the parathyroid gland, thus resulting in diagnostic errors, followed by inadequate treatment.

For the detection of parathyroid cells, we added a new marker—the expression level of the *GCM2* gene (glial cells missing homolog 2, a zinc finger transcription factor)—which is considered the main regulator of the development of parathyroid glands because they do not form in case of GCM2 deficiency. Notably, GCM2 expression is present both at the embryonic stage and after birth [26]. In our study, weak expression of the *GCM2* gene was detectable in some thyroid nodules, but the lowest level observed in the parathyroid gland was 150-fold greater than the maximum for the thyroid gland (the mean values differed 6400-fold).

This marker allows identification of nodules of the parathyroid gland but not determination of whether this lesion is benign or malignant. Adenomas of the parathyroid gland (80% of cases) and hyperplastic nodules of the parathyroid gland (15%) are the most prevalent lesions at this site, whereas carcinomas of the parathyroid gland are fairly rare (5%) [27]. In the present study, only PTA cases were analyzed; therefore, we cannot train the classifier to identify parathyroid carcinomas.

In our previous work [15], in comparison with other markers, the level of *HMGA2* expression was found to be the best single marker for the stratification of thyroid lesions and tumors into benign and malignant. In the present work, this result was confirmed; however, two trends transpired. Firstly, an increase in *HMGA2* expression occurs mainly in tumors originating from follicular cells: in all subtypes of PTC (*n* = 53; 84%), FTC (*n* = 10; 66.7%), and ATC (*n* = 8; 80%). In MTC in this work, an increase was registered only in 2 of 19 samples (10.5%), whereas in the previous work, it was 5.2%. In HCC and PTA, there was no increase in *HMGA2* expression. Secondly, an increase in *HMGA2* expression (above our chosen cutoff) was detectable in NIFTP (*n* = 4; 40%) and FTA (*n* = 8; 7.2%). This finding may indicate that although *HMGA2* expression increases with the progression of malignancy, this indicator does not always correlate with the degree of malignancy, i.e., this gene’s expression can be high in indolent tumors too.

Our study revealed that miR-146b expression is increased in PTC, miR-375 expression in MTC, and miR-221 expression in various carcinomas, especially in HCC. If we examine different PTC subtypes, then the expressions of miR-146b, -221, and -375 can be ranked from the highest to lowest by the median levels of relative expression as follows: TCV-PTC, CV-PTC, FV-PTC, SV-PTC, and NIFTP. The only deviation in this pattern was noted for miR-146b: its level was higher in CV-PTC, not in TCV-PTC. Notably, judging by the expression profile of these miRNAs, NIFTP was found to be closer to SV-PTC than to FV-PTC. A similar pattern was documented here for the prevalence of the V600E mutation of the *BRAF* gene: TCV-PTC (100%), CV-PTC (79.2%), FV-PTC (40.9%), SV-PTC (11.1%), and NIFTP (0%). Overall, this result is consistent with the published data on the occurrence of the *BRAF*^V600E^ mutation in different subtypes of PTC: TCV-PTC, 77–93% [28,29,30,31]; CV-PTC, 60–75.3%; FV-PTC, 10–40% [28,29,32,33]; and SV-PTC, 10% [34].

In the pattern described above, TCV-PTC features would logically be the most pronounced changes in molecular markers of malignancy. This is because this PTC subtype has aggressive clinical manifestations and a relatively high mortality rate. It becomes more prevalent with age and is more often associated with extrathyroid invasion (53.6%) and a reduction in 5-year survival compared with classic PTC (81.9% versus 97.8%, respectively) [31,35,36,37,38,39].

The finding that the profiles of molecular markers (*HMGA2*, miR-146b, -221, and -375) of SV-PTC are the closest to NIFTP is more difficult to explain. SV-PTC is defined as a distinct subtype of PTC by the WHO, and a recent meta-analysis of 11 studies with a total of 205 SV-PTC cases suggested that SV-PTC poses a higher risk of vascular invasion, tumor recurrence, and mortality in comparison with CV-PTC [34]. It is possible that SV-PTC has specific molecular markers that differ from those common for PTCs and, especially, CV-PTC cases.

In the clinical samples, we used the ratio of mtDNA to nDNA as a marker of Hürthle cells because they are distinguished by the accumulation of a large number of mitochondria [18]. In general, an increase in the number of mitochondria in tumor cells indicates a low proliferation rate [40], which is overall typical for most thyroid tumors, with the exception of low-differentiation and undifferentiated stages of this cancer. Our results match these data; for example, the average mtDNA/nDNA ratio for ATC is 464, 2084 for FTC, and 17,940 for HCC (Figure 4).

HCC (and HCA) is categorized by the WHO as a special type of tumor, among other reasons, owing to differences from FTC in the molecular profile and clinical behavior [41,42]. Our results imply that HCC may derive from FTC because mtDNA/nDNA values for FTC are on a continuum with the results for HCC (except for one sample that was identified as benign by our classifier). In other words, there were no outliers in the FTC group because they all fell into the HCC group (Figure 4). For other types of tumors, there were some outliers that corresponded to Hürthle cell subtypes of these tumors. For instance, for nine cases of FTA, the histological diagnosis indicated that they were Hürthle cell subtypes, and among them, in eight cases (88.9%), the mtDNA/nDNA ratio exceeded the cutoff specified in our classifier.

The molecular profile of HCC (*HMGA2*, miR-146b, -221, and -375) differed slightly from FTC, mainly by the increased expression of miR-221 (*p* = 5.32 × 10^−4^) and lowered expression of *HMGA2* (*p* = 1.24 × 10^−3^). In this study, none of the FTC samples showed miR-221 overexpression, although such examples have previously been reported by us [15]. Nevertheless, increased expression of miR-221, which is not typical for FTC in general, is one of the main features of HCC (Figure 3). This factor probably determines the more aggressive behavior of HCC; this hypothesis applies if we extrapolate the PTC data to HCC [43].

For the results of typing of thyroid tumors and lesions, most discrepancies between molecular and histological diagnoses were found in the NIFTP and FTC groups. For instance, 60% of NIFTP samples were categorized as benign by the molecular classifier; this pattern is to be expected because NIFTP belongs to the category of borderline tumors. Nevertheless, this finding should be regarded as an error because these tumors should be resected despite their indeterminate malignancy status. For FTC, 33% of the samples fell into the benign group, and four of the five incorrectly identified FTCs were minimally invasive.

These discrepancies are probably related primarily to, firstly, the molecular classifier not being initially trained on samples with NIFTP; secondly, the FN-MM group, to which FTC belongs, is the only group without specific markers as revealed by our results and is basically a collection of samples that did not fall into groups PTC, MTC, and HCC. Nonetheless, the histological criteria for NIFTP and other neoplasms with an indeterminate malignant potential are characterized by low reproducibility of results among pathologists [44,45]. An accurate assessment of capsular and vascular invasion (necessary for the differential diagnosis of FTC) is often impossible due to artifacts arising from tissue processing and, even more so, due to incomplete histological examination of the capsule of follicular neoplasms, especially large ones.

The results of our study showed the high specificity of the classifier for detecting all major types of thyroid carcinoma. Only the sensitivity of FTC detection was low (66.7% versus 93–100% for other types of carcinoma); in other cases, the use of specific markers helped to achieve high accuracy: MTC, 100%; PTC, 98.4%; and HCC, 99.6%.

If we consider PTC in terms of histological subtypes, then 100% of the samples histologically assigned to CV-PTC, TCV-PTC, and FV-PTC were categorized by the molecular classifier as papillary carcinoma, whereas histologically diagnosed SV-PTC was assigned to papillary carcinoma only in 55.5% of cases. Other 22.25% of histologically identified SV-PTC samples fell into the FN-MM group and 22.25% into the benign group. Thus, molecular markers of papillary carcinoma are not sufficient for accurate molecular identification of this PTC variant. Evidently, a search for additional markers is required. Additional markers are also necessary for the differential diagnosis of ATC. Originally, our molecular classifier was not trained on ATC samples; nevertheless, all the studied cases were classified as malignant tumors (in 80% cases as PTC and in 20% cases as FN-MM).

Comparing the results obtained in this work with our previous study where only samples from the group of indeterminate cytological diagnoses (Bethesda III + IV) [23] were analyzed, we note a satisfactory match between the obtained diagnostic characteristics. This should help to solve the problem of detecting signs of malignant tumors (Table 3).

As shown in Table 3, when ATC and NIFTP were excluded from the analyzed group, higher sensitivity and PPV for detecting cancer were obtained in this study, but this accomplishment is attributable to a large proportion of clinical samples (41.5% versus 30% in the previous study) of carcinomas in the study population and to the inclusion of numerous clinical samples of CV-PTC and TCV-PTC. Accordingly, in the previous study, specificity and NPV were higher, which can be explained by a higher proportion of benign samples (69.7% versus 58.5% in the present study). Nonetheless, specificity and NPV are only slightly lower in the current study. In particular, NPV remained within 95%, which is considered necessary for a test that can obviate surgical treatment [46]. A somewhat different picture emerges if we include ATC and NIFTP specimens in the analyzed group when regarding NIFTP as a carcinoma. As described above, all ATC specimens were identified as carcinomas, but only 40% of NIFTP specimens were identified as carcinomas; therefore, specificity did not change and PPV slightly increased, but sensitivity and NPV diminished noticeably.

In a real-world patient population, the presence of NIFTP cases should not much affect PPV and NPV because although sensitivity and specificity characterize a test independently of the incidence of cancer (in this case, in the study population), but NPV and PPV depend on the incidence. Assuming fixed sensitivity and specificity, Bayes’ theorem can predict the dependence of a test’s NPV and PPV on the incidence of cancer, and even in the general population [47]. These dependences are illustrated in Figure 6; the curves of functions are not shown in Figure 6 if specificity or sensitivity was 100% (e.g., for our MTC cases).

If we consider a cancer incidence rate of 30%, as was the case in the previous work in the group of indeterminate cytological diagnoses, then for the detection of malignant tumors, PPV is 82.8% and NPV is 96.9%. If ATC and NIFTP are included in the calculations, then PPV is 82.2% and NPV is 95.3%.

Some limitations of this study should be mentioned. Firstly, in some groups, the sample size was low (for example, *n* = 6 for HCC), which could affect the obtained diagnostic characteristics for the detection of these nosological entities. Secondly, the entire study population was from one medical center, and third, we did not include cases of PDTC, which are often categorized as an indeterminate pathology by cytological examination, for example, as a follicular tumor.

## 4. Materials and Methods

### 4.1. Selection of Clinical Samples

This retrospective study is based on clinical samples obtained from patients who underwent fine-needle aspiration biopsy at St. Petersburg State University N.I. Pirogov Clinic of High Medical Technologies, St. Petersburg, Russia. Cytological and histological analyses of the material were conducted at the National Center of Clinical Morphological Diagnostics (St. Petersburg, Russia).

The study protocol was approved by an independent ethics committee, Arte Med Assistance, LLC. The clinical samples were obtained in accordance with the current laws and regulations of the Russian Federation, and informed consent was obtained from each patient. All data were depersonalized.

The eligibility criteria were as follows: one cytological slide per patient, availability of a cytological report and histological report, age over 18 years, the size of a nodule according to ultrasonographic examination > 5 mm, and a documented comparison between the ultrasonographic description of the thyroid nodule at preoperative diagnosis and a macroscopic description of this thyroid nodule in the operation documents. A sample was included in the study only if the cytological report and other characteristics, such as size and location, were consistent with the nodule description in the final histological report. In case of doubt regarding the consistency of nodules between the cytological and histological reports due to the presence of more than one nodule in the patient or a lack of details in the reports, the sample was excluded from the study.

At the first stage, cytological samples prepared by a standard procedure were examined. The cytological preparations were fixed and stained with azure-eosin dye following the May–Grunwald–Giemsa method. The cellular composition of the thyroid nodule aspirates was assessed in accordance with the criteria of the international classification adopted in 2017 for assessing thyroid nodules and standardizing the results of classification (The Bethesda System for Reporting Thyroid Cytopathology (TBSRTC), Second Edition, 2017) for each diagnostic group.

A retrospective analysis was also performed on the histological preparations of nodules. All histological samples after thyroidectomy or lobectomy were prepared by the standard method from tissue material fixed in 10% buffered formalin, followed by routine staining with hematoxylin and eosin. The pathological characteristics were evaluated in accordance with the criteria of the Classification of Tumors of the Thyroid Gland, adopted by the WHO in 2017: WHO Classification of Tumors of Endocrine Organs.

For NIFTP, we used revised 2018 criteria [48]: (1) encapsulation or clear demarcation; (2) a follicular growth pattern with no well-formed papillae, no psammoma bodies, with a <30% solid, trabecular, or insular growth pattern; (3) nuclear score 2–3; (4) no vascular or capsular invasion; (5) no tumor necrosis; (6) the absence of high mitotic activity (<3 mitoses per 10 high-power fields). In all NIFTP cases, the nodule was solitary, and no multifocal pattern was found; the nodules either had a thick/thin or partial capsule or were clearly demarcated from the adjacent thyroid parenchyma.

In the goiter group, we included patients with thyroid non-neoplastic lesions: diffuse toxic hyperplasia, nodular/multinodular hyperplasia, adenomatous hyperplasia, and/or chronic lymphocytic thyroiditis (1 case).

The characteristics of the samples are shown in Table 4.

Thus, 278 cytological samples from 278 patients were analyzed in this work. The distribution of pathologies was as follows: goiter, 34 cases; FTA, 111; FTC, 15; HCC, 6; CV-PTC, 24; FV-PTC, 21; TCV-PTC, 8; SV-PTC, 9; MTC, 19; PTA, 11; NIFTP, 10; and ATC, 10 cases. There were 60 men (21.6% of cases), mean age 49 (18–73) years, and 218 women (78.4% of cases), mean age 51 (18–85) years. Patient data are given in Appendix A.

### 4.2. Total Nucleic Acid Extraction

The nucleic acids were extracted from fine-needle aspiration cytological preparations as previously described [49]: Each dried cytological preparation was washed in a microcentrifuge tube with three 200 μL portions of guanidine lysis buffer. The sample was vigorously mixed and incubated in a thermal shaker for 15 min at 65 °C. Next, an equal volume of isopropanol was added. The reaction solution was thoroughly mixed and kept at room temperature for 5 min. After centrifugation for 10 min at 14,000× *g*, the supernatant was discarded, and the pellet was washed with 500 μL of 70% ethanol and 300 μL of acetone. Finally, the RNA was dissolved in 200 μL of deionized water. If not analyzed immediately, RNA samples were stored at −20 °C until further use.

### 4.3. Molecular Analysis

The assessment of relative levels of *HMGA2* and *GCM2* mRNA expression (normalized to household gene *PGK1* [50]); of miR-146b, -221, and -375 levels (normalized to the geometric mean level of miR-29b, -23a, and -197); and of the ratio of mtDNA to nDNA, as well as the detection of somatic mutation V600E in the *BRAF* gene, was carried out as described earlier [15]. The oligonucleotides for detecting *GCM2* mRNA were as follows: TaqMan probe (SIMA)-TGC+CTCAGGAGCTGGCCCTC-(BHQ1) [+C=C-LNA], forward primer CTCAGCTGGGACATCAAC, and reverse primer (and primer for reverse transcription) AGGCCACTCTCGGAATTG.

### 4.4. Data Analysis

The data were processed in Excel (Microsoft, Redmond, WA, USA) or Statistica 9.1 (StatSoft Inc., Palo Alto, CA, USA). Diagnostic characteristics were determined by means of standard 2 × 2 contingency tables comparing qualitative, binary molecular test results (positive or negative) relative to the reference standard diagnoses made by histopathological examination. The confidence intervals for sensitivity, specificity, and accuracy were calculated with the Clopper–Pearson method. The confidence intervals for the predictive values are the logit confidence intervals given in [51]. Two independent samples were compared by quantitative traits via the Mann–Whitney U test.

## 5. Conclusions

We demonstrated that the diagnostic panel including the analysis of miRNA expression (miR-146b, -221, and -375), mRNA expression (*HMGA2* and *GCM2*), the V600E mutation in the *BRAF* gene, and the ratio of mtDNA to nDNA enables accurate identification of parathyroid and thyroid tumors: PTC, MTC, HCC, and, with lower accuracy, malignant follicular tumors (PPV was 43.5% for the latter).

## Figures and Tables

**Figure 1 cancers-13-00237-f001:**
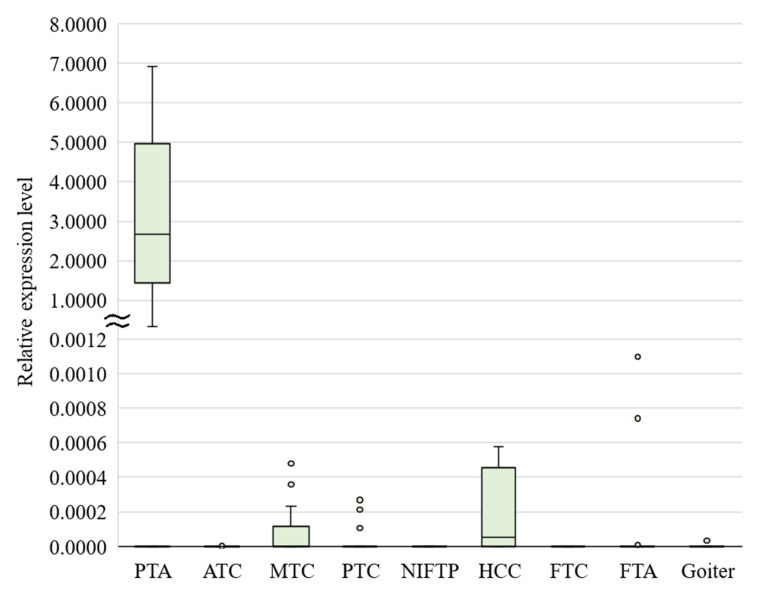
*GCM2* mRNA expression in parathyroid and thyroid tumors and goiters, presenting the median value, upper and lower quartiles, a nonoutlier range, and outliers (circles). PTA, parathyroid adenoma; ATC, anaplastic thyroid carcinoma; MTC, medullary thyroid carcinoma; PTC, papillary thyroid carcinoma; NIFTP, noninvasive follicular thyroid neoplasm with papillary-like nuclear features; HCC, Hürthle cell carcinoma; FTC, follicular thyroid carcinoma; FTA, follicular thyroid adenoma.

**Figure 2 cancers-13-00237-f002:**
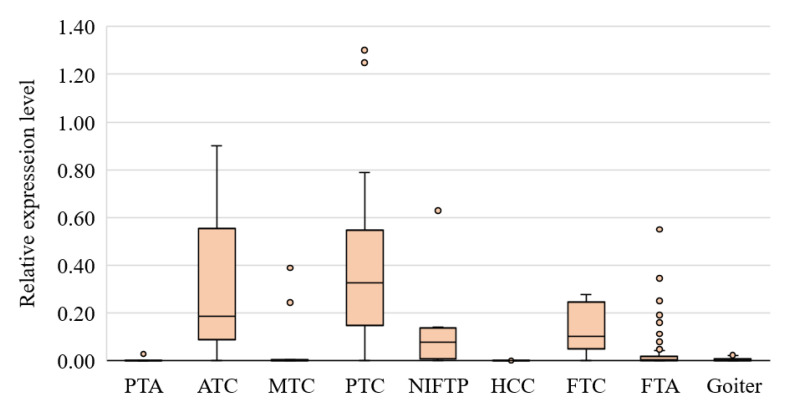
*HMGA2* mRNA expression in goiters and thyroid/parathyroid tumors, presenting the median value, upper and lower quartiles, a nonoutlier range, and outliers (circles).

**Figure 3 cancers-13-00237-f003:**
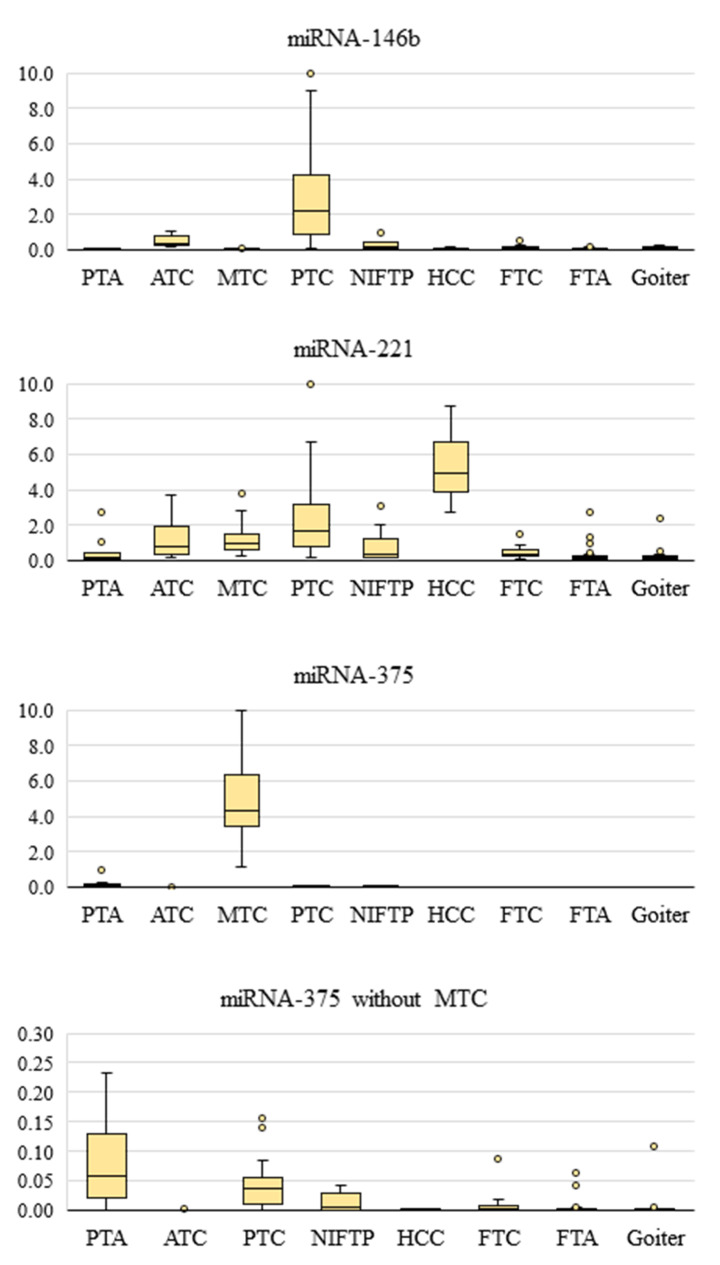
Relative expression levels of three miRNAs in goiters and thyroid/parathyroid tumors. The data are normalized; the figure shows the median value, upper and lower quartiles, a nonoutlier range, and outliers (circles).

**Figure 4 cancers-13-00237-f004:**
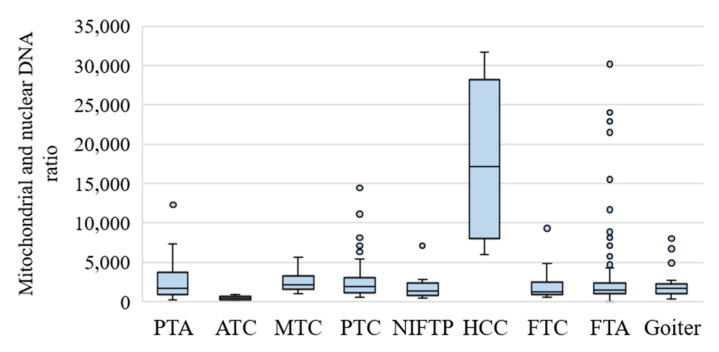
The mtDNA/nDNA ratio in goiters and thyroid/parathyroid tumors. The figure presents the median value, upper and lower quartiles, a nonoutlier range, and outliers (circles).

**Figure 5 cancers-13-00237-f005:**
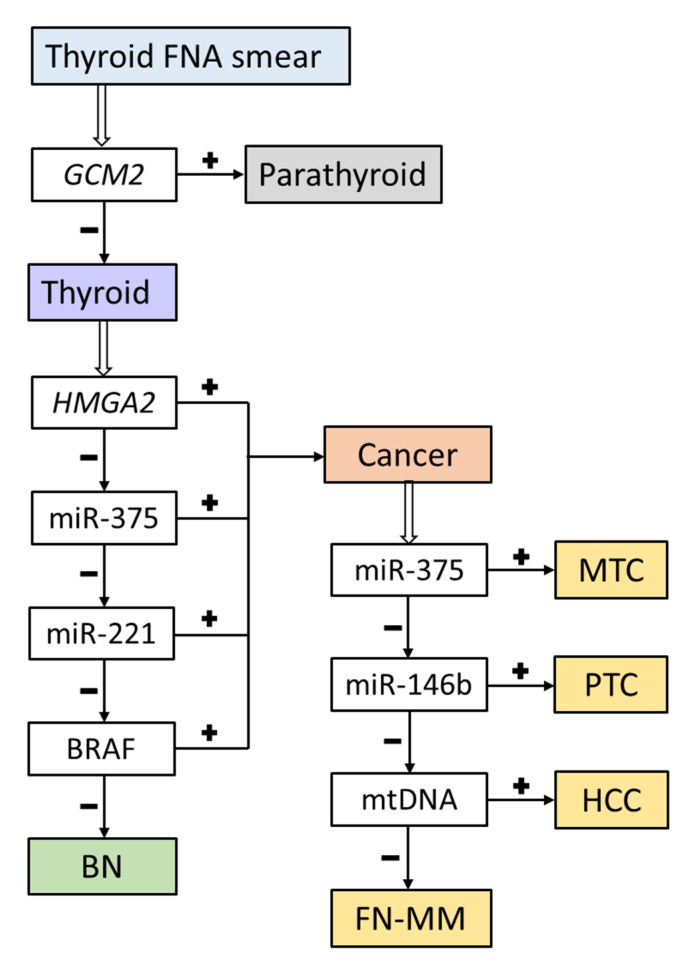
The decision tree for classifying samples into benign and malignant followed by cancer typing. +/−: exceeding the chosen cutoff or identifying the *BRAF*^V600E^ mutation/not exceeding the chosen cutoff or not identifying the *BRAF*^V600E^ mutation, BN: benign nodule, MTC: medullary thyroid carcinoma, PTC: papillary thyroid carcinoma, HCC: Hürthle cell thyroid carcinoma, FN-MM: follicular neoplasm with markers of malignancy.

**Figure 6 cancers-13-00237-f006:**
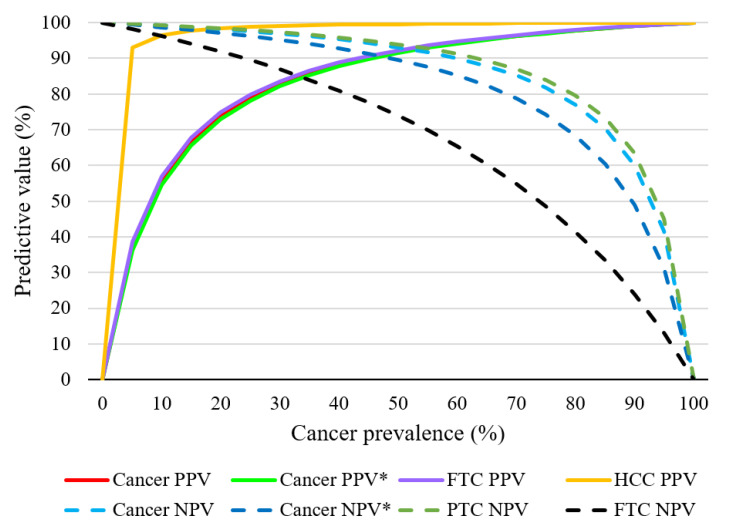
Expected PPV and NPV according to the sensitivity and specificity observed in the present study versus thyroid cancer prevalence in a given study population. Cancer PPV*/NPV*: with the addition of ATC and NIFTP specimens into the analyzed group.

**Table 1 cancers-13-00237-t001:** Molecular versus histological classification of thyroid nodules.

Histological Diagnosis (Reference Standard)	Molecular Classification Results				
	MTC, n	PTC, n	FN-MM, n	HCC, n	Benign, n
MTC (n = 19)	19				
PTC all variants (n = 63)		59	2		2
TCV-PTC (n = 8)		8			
CV-PTC (n *=* 24)		24			
FV-PTC (n = 22)		22			
SV-PTC (n = 9)		5	2		2
HCC (n = 6)				6	
FTC (n = 15)			10		5
Goiter (n = 34)			1		33
FTA (n = 111)			10	1	100

**Table 2 cancers-13-00237-t002:** Diagnostic characteristics of the molecular classifier for malignant tumors (with 95% confidence intervals). PPV: positive predictive value, NPV: negative predictive value.

	Cancer, n = 103	MTC, n = 19	PTC, n = 63	FTC, n = 15	HCC, n = 6
Specificity, %	91.7 (86.0–95.6)	100.0 (98.4–100.0)	100.0 (98.0–100.0)	94.4 (90.6–97.0)	99.6 (97.7–100.0)
Sensitivity, %	93.2 (86.5–97.2)	100.0 (82.3–100.0)	93.6 (84.5–98.2)	66.7 (38.4–88.2)	100.0 (54.1–100.0)
Accuracy, %	92.3 (88.3–95.3)	100.0 (98.5–100.0)	98.4 (95.9–99.6)	92.7 (88.8–95.6)	99.6 (97.8–100.0)
PPV, %	88.9 (82.3–93.2)	100.0	100.0	43.5 (28.9–59.3)	85.7 (45.9–97.7)
NPV, %	95.0 (90.3–97.5)	100.0	97.9 (94.7–99.2)	97.8 (95.5–98.9)	100.0

**Table 3 cancers-13-00237-t003:** Diagnostic characteristics of the molecular classifiers for malignant tumors (with a 95% confidence interval) from the current and previous studies.

	Bethesda III-IV (n = 122)	Bethesda II-VI (n = 248) Current Study	Bethesda II-VI (n = 268) * Current Study
Specificity, %	92.9 (85.3–97.4)	91.7 (86.0–95.6)	91.7 (86.0–95.6)
Sensitivity, %	89.2 (74.6–97.0)	93.2 (86.5–97.2)	89.4 (82.6–94.2)
Accuracy, %	91.8 (85.4–96.0)	92.3 (88.3–95.3)	90.7 (86.5–93.9)
PPV, %	84.6 (71.6–92.3)	88.9 (82.3–93.2)	90.2 (84.2–94.0)
NPV, %	95.2 (88.6–98.0)	95.0 (90.3–97.5)	91.1 (85.9–94.5)

* ATC and NIFTP specimens were added to the analyzed group; NIFTP was regarded as a carcinoma because it requires resection.

**Table 4 cancers-13-00237-t004:** Samples used in this work. TBSRTC: The Bethesda System for Reporting Thyroid Cytopathology.

	Histological Diagnosis (Abbreviated)	ICD-O Code According to WHO Classification, 2017	Number of Samples	Results of Cytological Examination According to TBSRTC, 2017
1	Follicular thyroid adenoma	8330/0, 8290/0	111	Bethesda IV (111 samples)
2	Follicular thyroid carcinoma	8335/3, 8339/3	15	Bethesda IV (15 samples)
3	Hürthle cell carcinoma	8290/3	6	Bethesda IV (6 samples)
4	Follicular variant of papillary thyroid carcinoma (FV-PTC)	8340/3	22	Bethesda IV (8 samples), Bethesda VI (14 samples)
5	Papillary thyroid carcinoma, variants other than FV-PTC	8260/3, 8343/3, 8341/3, 8344/3, 8342/3	41	Bethesda IV (6 samples), Bethesda VI (35 samples)
6	NIFTP	8340/3	10	Bethesda IV (10 samples)
7	Medullary thyroid carcinoma	8345/3	19	Bethesda IV (2 samples), Bethesda V (3 samples), Bethesda VI (14 samples)
8	Anaplastic thyroid carcinoma	8020/3	10	Bethesda VI (10 samples)
9	Parathyroid adenoma	8140/0	10	Bethesda IV (10 samples)
10	Goiter		34	Bethesda I (2 samples), Bethesda II (25 samples), Bethesda IV (6 samples), Bethesda V (1 sample).
	Total number of samples		278	Bethesda I (2 samples), Bethesda II (25 samples), Bethesda IV (174 samples), Bethesda V (4 samples), Bethesda VI (73 samples)

## Data Availability

The data presented in this study are available on request from the corresponding author.

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
