# Peer review of "Preoperative Typing of Thyroid and Parathyroid Tumors with a Combined Molecular Classifier"

_cancers, 2021, doi:10.3390/cancers13020237_

Round 1

Reviewer 1 Report

The authors addressed the critical point.

Author Response

Thanks for the review.

Reviewer 2 Report

There are no further comments re. scientific contents, which now deemed acceptable.   I requested professional proofreading from the authors and they claimed "We had the manuscript edited by a suitable company, and the certificate is enclosed" However, the quality of English language remains poor.

This was a major comment and this manuscript shouldn't go without a good quality language proofreading.

Author Response

The manuscript has undergone English editing via MDPI service. I hope this will be enough.

This manuscript is a resubmission of an earlier submission. The following is a list of the peer review reports and author responses from that submission.

Round 1

Reviewer 1 Report

The authors describe a molecular classifier - diagnostic panel comprising analysis of miRNA expression, mRNA expression, BRAF V600E mutation, and the 32 mitochondrial-to-nuclear DNA ratio to distinguish the various forms of thyroid tumors, including benign adenomas and all types of thyroid cancers (except for poorly differentiated papillary thyroid carcinoma), and they also add an additional preliminary molecular diagnostic step of distinguishing parathyroid tumors from thyroid tumors.

Overall, the manuscript is organized well, and the data are well-explained. The following aspects need attention:

  1. The abstract does not mention anything regarding parathyroid tumor evaluation performed in the study. A sentence or two should be mentioned about molecular diagnostics of the parathyroid tumors that were performed in this study, especially given that the term parathyroid tumors is mentioned in the title.
  2. In lines 97 and 98, the abbreviations PTC and ATC could be used.
  3. For the sentence on lines 107 – 109, please provide the reference.
  4. The term ‘goiter’ is utilized throughout the manuscript, including in several figures and tables. However, goiter is more of a clinical term rather than a histopathological diagnosis. It would be useful if the authors define what exactly qualified as ‘goiter’ in their study (do the authors mean hyperplastic/adenomatoid nodules?), and they need to clarify why goiter is represented as a separate entity along with follicular adenomas, and the various thyroid cancers, and if they are hyperplastic/adenomatoid nodules, then consider replacing the term 'goiter' with 'hyperplastic nodule' or 'adenomatoid nodule'.
  5. In figure 8, among the excluded participants, what does ‘slides from one patient’ mean? Does this mean duplicates?
  6. Line 360, the word ‘of’ is missing next to ‘this PTC variant’.
  7. In lines 371 – 374, the authors mention the following: “It is worth mentioning that in this study, higher sensitivity and PPV for detecting cancer were obtained, but this accomplishment is apparently attributable to a large proportion of clinical samples (41.5% versus 30% in the previous study) of carcinomas in the study population and to the inclusion of numerous clinical samples of cPTC and TCV-PTC.” Here the authors are comparing the biopsy samples from their current study (which are Bethesda II to VI) to those from their prior study (which were Bethesda III and IV). So, apart from the larger proportion of carcinomas in the current study, wouldn’t the presence of more benign lesions (Bethesda II) also affect the diagnostic characteristics mentioned in table 3? It would be useful for the authors to clarify this.

Author Response

Review Report 1

The authors describe a molecular classifier - diagnostic panel comprising analysis of miRNA expression, mRNA expression, BRAF V600E mutation, and the 32 mitochondrial-to-nuclear DNA ratio to distinguish the various forms of thyroid tumors, including benign adenomas and all types of thyroid cancers (except for poorly differentiated papillary thyroid carcinoma), and they also add an additional preliminary molecular diagnostic step of distinguishing parathyroid tumors from thyroid tumors.

Overall, the manuscript is organized well, and the data are well-explained. The following aspects need attention:

  1. The abstract does not mention anything regarding parathyroid tumor evaluation performed in the study. A sentence or two should be mentioned about molecular diagnostics of the parathyroid tumors that were performed in this study, especially given that the term parathyroid tumors is mentioned in the title.

We have moved from the Results to Introduction two paragraphs that describe why we wanted to distinguish between the parathyroid gland and thyroid gland, and with which marker we were going to do this (lines 106–113).

  1. In lines 97 and 98, the abbreviations PTC and ATC could be used.

We have replaced this text with the abbreviations.

  1. For the sentence on lines 107 – 109, please provide the reference.

The reference is now inserted (line 116).

  1. The term ‘goiter’ is utilized throughout the manuscript, including in several figures and tables. However, goiter is more of a clinical term rather than a histopathological diagnosis. It would be useful if the authors define what exactly qualified as ‘goiter’ in their study (do the authors mean hyperplastic/adenomatoid nodules?), and they need to clarify why goiter is represented as a separate entity along with follicular adenomas, and the various thyroid cancers, and if they are hyperplastic/adenomatoid nodules, then consider replacing the term 'goiter' with 'hyperplastic nodule' or 'adenomatoid nodule'.

The definition was added to the Materials and Methods (lines 445-448): “In the goiter group, we included patients with benign nontumorous alterations of the thyroid: diffuse toxic hyperplasia, nodular/multinodular hyperplasia, adenomatoid nodule, adenomatous hyperplasia, and/or chronic lymphocytic thyroiditis (1 case).”

  1. In figure 8, among the excluded participants, what does ‘slides from one patient’ mean? Does this mean duplicates?

This means that when several cytological slides from one patient were available, we kept one while the others were excluded from the study.

  1. Line 360, the word ‘of’ is missing next to ‘this PTC variant’.

Thanks for noticing; “of” has been inserted.

  1. In lines 371 – 374, the authors mention the following: “It is worth mentioning that in this study, higher sensitivity and PPV for detecting cancer were obtained, but this accomplishment is apparently attributable to a large proportion of clinical samples (41.5% versus 30% in the previous study) of carcinomas in the study population and to the inclusion of numerous clinical samples of cPTC and TCV-PTC.” Here the authors are comparing the biopsy samples from their current study (which are Bethesda II to VI) to those from their prior study (which were Bethesda III and IV). So, apart from the larger proportion of carcinomas in the current study, wouldn’t the presence of more benign lesions (Bethesda II) also affect the diagnostic characteristics mentioned in table 3? It would be useful for the authors to clarify this.

We have inserted such a sentence into the text (lines 379–381); however, it should be noted that Figure 10 deals with the relation between the prevalence of cancer and NPV/PPV.

Reviewer 2 Report

Authors present an interesting study on molecular classifiers differentiating thyroid tumors and also parathyroid tumors. It's strong value is the inclusion of NIFTP tumors. Comments: Authors do not specify the criteria of pathological diagnosis of NIFTP. This lack deserves an attention, because the diagnositc criteria of NIFTP are still changing.

Author Response

Review Report 2

Authors present an interesting study on molecular classifiers differentiating thyroid tumors and also parathyroid tumors. It's strong value is the inclusion of NIFTP tumors. Comments: Authors do not specify the criteria of pathological diagnosis of NIFTP. This lack deserves an attention, because the diagnositc criteria of NIFTP are still changing.

NIFTP diagnostic criteria have been added to Materials and Methods (lines 441-445): (1) encapsulation or clear demarcation; (2) a follicular growth pattern with all of the following: <1% papillae, no psammoma bodies, <30% solid, trabecular, or insular growth pattern; (3) nuclear features of papillary carcinoma; (4) no lympho-vascular or capsular invasion; (5) no tumor necrosis; (6) the absence of high mitotic activity (<3 mitoses per 10 high-power fields).

Reviewer 3 Report

In this manuscript, authors report the performance of a molecular test evaluating the gene expression levels of two genes and three miRNA, the mithocondrial/nuclear DNA ratio and the BRAF V600E mutation on 278 thyroid nodules samples with known histology. Tumor histotypes included medullary thyroid carcinoma, papillary thyroid carcinoma, Hurthle cell carcinoma, follicular thyroid carcinoma, NIFTP, anaplastic cancer, nodular goiter and follicular adenoma samples.

The main limitations of this study are the following:

  • The relationship between molecular analysis and cytology is poorly considered. The study loses track of the purpose that a molecular test should have in this context. The HCC-specific markers included in the proposed panel (mir-221 expression and mitochondrial/nuclear DNA ratio) are intended to identify Hurthle cell neoplasms. However, the diagnostic issue on cytology for this class of tumors is not represented by the identification of oncocytic cells, but it stands in the differential diagnosis between benign and malignant lesions. Similarly, the identification of MTC on cytology is rarely challenging; the diagnosis of anaplastic cancer does not require a molecular classifier, while it would have been interesting to include poorly differentiated carcinomas. The parathyroid cells marker (GCM2 expression) could be considered useful, but in this cohort the issue of the challenging diagnosis between parathyroid adenoma and carcinoma is not treated, since parathyroid carcinoma samples were not analyzed. The same can be said for the absence of Hurthle cell adenoma samples.

Finally, the complete lack of Bethesda III nodules represents one of the main limitations of this work.

  • Lack of a validation cohort. This is a highly selected cohort, enriched in malignant tumors and BRAF-positive PTC variants. Specificity, sensitivity and predictive values of the “molecular classifier” are then calculated on a series that did not represent a real-life indeterminate cohort. Finally, besides the lack of consideration for some crucial type of lesions, the issue of overfitting must be considered in this series since there are almost less predictors than categories to identify. I suggest then that a validation cohort is necessary to confirm these results.

Minor comments:

  • Line 316: It is stated that, according to the WHO indications, HCC is a variant of FTC. This is not correct.
  • Lines 334-336: “60% of NIFTP samples were categorized as benign by the molecular classifier, which is rather a positive result, given that NIFTP is considered an indolent tumor…” A benign call for NIFTP should not be considered as a true negative result, as they need surgical removal according to the current pathological and clinical guidelines.
  • Lines 347-350: the hypothesis that the follicular adenoma samples classified as malignant could have been RAS-positive should be experimentally confirmed.
  • Table 3: The comparison of the diagnostic performance of the molecular test between the previous study, which included Bethesda III and IV nodules only, and the present one (Bethesda II, IV, V, VI nodules) is in my opinion inappropriate.

Author Response

Review Report 3

In this manuscript, authors report the performance of a molecular test evaluating the gene expression levels of two genes and three miRNA, the mithocondrial/nuclear DNA ratio and the BRAF V600E mutation on 278 thyroid nodules samples with known histology. Tumor histotypes included medullary thyroid carcinoma, papillary thyroid carcinoma, Hurthle cell carcinoma, follicular thyroid carcinoma, NIFTP, anaplastic cancer, nodular goiter and follicular adenoma samples.

The main limitations of this study are the following:

  • The relationship between molecular analysis and cytology is poorly considered. The study loses track of the purpose that a molecular test should have in this context. The HCC-specific markers included in the proposed panel (mir-221 expression and mitochondrial/nuclear DNA ratio) are intended to identify Hurthle cell neoplasms. However, the diagnostic issue on cytology for this class of tumors is not represented by the identification of oncocytic cells, but it stands in the differential diagnosis between benign and malignant lesions.
  • Perhaps, the text was not very clear, but only the mtDNA/nDNA ratio is a marker of oncocytes, whereas by means of miR-221, it is possible to distinguish malignant oncocytic tumors from benign ones.

Similarly, the identification of MTC on cytology is rarely challenging; the diagnosis of anaplastic cancer does not require a molecular classifier, while it would have been interesting to include poorly differentiated carcinomas.

We agree, but the main purpose of the molecular classifier is to determine whether there is cancer in Bethesda III-IV samples, whereas determining the tumor type is a supplementary aim that did not require inclusion of a large number of additional markers. Because there was an opportunity for subtyping, we wanted to test its accuracy, even if this does not apply to all types of tumors for which identification is worthwhile.

The parathyroid cells marker (GCM2 expression) could be considered useful, but in this cohort the issue of the challenging diagnosis between parathyroid adenoma and carcinoma is not treated, since parathyroid carcinoma samples were not analyzed.

For the time being, the main aim is to distinguish between parathyroid tumors and thyroid tumors. We are planning to work on distinguishing malignant parathyroid tumors from benign ones after collecting a sufficient number of clinical samples.

The same can be said for the absence of Hurthle cell adenoma samples.

Such samples were present in the analysis but were not examined separately from other adenomas because their separate identification is not being planned yet. They are mentioned in the Discussion, lines 328–330.

Finally, the complete lack of Bethesda III nodules represents one of the main limitations of this work.

Sorry, this work is not focused on cytological diagnoses; this topic was addressed in our previous study: Titov, S.; Demenkov, P.S.; Lukyanov, S.A.; Sergiyko, S.V.; Katanyan, G.A.; Veryaskina, Y.A.; Ivanov, M. K. Preoperative detection of malignancy in fine-needle aspiration cytology (FNAC) smears with indeterminate cytology (Bethesda III, IV) by a combined molecular classifier. J Clin Pathol 2020, 73(11), 722–727.

  • Lack of a validation cohort. This is a highly selected cohort, enriched in malignant tumors and BRAF-positive PTC variants. Specificity, sensitivity and predictive values of the “molecular classifier” are then calculated on a series that did not represent a real-life indeterminate cohort. Finally, besides the lack of consideration for some crucial type of lesions, the issue of overfitting must be considered in this series since there are almost less predictors than categories to identify. I suggest then that a validation cohort is necessary to confirm these results.
  • In its essence, this work is a validation study; therefore, one could say that the study population here is a validation cohort for the previous work where we constructed the molecular classifier: Titov, S.E.; Ivanov, M.K.; Demenkov, P.S.; Katanyan, G.A.; Kozorezova, E.S.; Malek, A.V.; Veryaskina, Y.A.; Zhimulev, I.F. Combined quantitation of HMGA2 mRNA, microRNAs, and mitochondrial-DNA content enables the identification and typing of thyroid tumors in fine-needle aspiration smears. BMC cancer 2019, 19(1), 1010.

In the study “Preoperative detection of malignancy in fine-needle aspiration cytology (FNAC) smears with indeterminate cytology (Bethesda III, IV) by a combined molecular classifier,“ we validated the classifier on a study population with indeterminate cytological diagnoses regarding the ability to distinguish benign from malignant tumors. In the present work, we validated the ability of the classifier to subtype malignant tumors. All these studies were performed on 3 independent study populations.

Minor comments:

  • Line 316: It is stated that, according to the WHO indications, HCC is a variant of FTC. This is not correct. We agree that this is an inaccuracy; the sentence has been replaced.
  • Lines 334-336: “60% of NIFTP samples were categorized as benign by the molecular classifier, which is rather a positive result, given that NIFTP is considered an indolent tumor…” A benign call for NIFTP should not be considered as a true negative result, as they need surgical removal according to the current pathological and clinical guidelines. The sentence has been rephrased.
  • Lines 347-350: the hypothesis that the follicular adenoma samples classified as malignant could have been RAS-positive should be experimentally confirmed. Indeed, this is a speculative theory; we have deleted the paragraph in question.
  • Table 3: The comparison of the diagnostic performance of the molecular test between the previous study, which included Bethesda III and IV nodules only, and the present one (Bethesda II, IV, V, VI nodules) is in my opinion inappropriate. The comparison of the results obtained on different study populations may not be entirely correct but is interesting for determining whether the results are consistent. Possible problems with this kind of comparison are discussed in the text (lines 376-381).

Reviewer 4 Report

To the Authors:

The work entitled “Preoperative typing of the thyroid and parathyroid tumors with a combined molecular classifier,” presented by Sergei E. Titov et al., improves a published algorithm necessary to detect the thyroid cancer subtypes.

Minor revisions:

The molecular analysis is well performed. For this reason, I suggest one minor revision and a question:

-          Figure 1 can be represented as only one graph with a cut (with two different scales) on the Y-axis.

-          Can the authors explain the choice of gene/miRNAs used in the real-time data normalization? In particular, the household gene PGK1 and the miR-29b, -23a, and -197.

Major revisions:

Does the statistical analysis show a very high specificity of the miRNA chosen for the analysis, are included all samples on the specificity calculation? Understanding the pipeline's real efficacy, it is necessary to include all samples to have a global vision of the results.

Example: among the 70 PTC analyzed (flow through this study, figure 8), 59 were real PTC at the final diagnosis. The specificity is lower than 100%.

A revision of the statistical analysis is mandatory.

Author Response

Review Report 4

The work entitled “Preoperative typing of the thyroid and parathyroid tumors with a combined molecular classifier,” presented by Sergei E. Titov et al., improves a published algorithm necessary to detect the thyroid cancer subtypes.

Minor revisions:

The molecular analysis is well performed. For this reason, I suggest one minor revision and a question:

-          Figure 1 can be represented as only one graph with a cut (with two different scales) on the Y-axis.

Sorry, we were unable to figure out how to do this in Excel.

-          Can the authors explain the choice of gene/miRNAs used in the real-time data normalization?

In particular, the household gene PGK1 and the miR-29b, -23a, and -197. The use of the PGK1 gene as a normalizer for HMGA2 comes from the following study: Lappinga PJ, Kip NS, Jin L, Lloyd RV, Henry MR, Zhang J, Nassar A. HMGA2 gene expression analysis performed on cytologic smears to distinguish benign from malignant thyroid nodules. Cancer Cytopathol. 2010;118:287-97. This reference has been inserted into the manuscript (line 471).

As for miRNA, we chose the normalizers ourselves (on the basis of our own data) by means of the GeNorm software: Vandesompele, J., De Preter, K., Pattyn, F. et al. Accurate normalization of real-time quantitative RT-PCR data by geometric averaging of multiple internal control genes. Genome Biol 3, research0034.1 (2002).

Major revisions:

Does the statistical analysis show a very high specificity of the miRNA chosen for the analysis, are included all samples on the specificity calculation? Understanding the pipeline's real efficacy, it is necessary to include all samples to have a global vision of the results.

Indeed, not all specimens were included in the calculations; ATC and NIFTP were excluded because the classifier is not designed to identify these types of tumors, and they would simply increase the error, decreasing specificity and PPV for some diagnoses. The idea was to determine "pure" subtyping accuracy for the subtypes of tumors on which the classifier has been trained. The obtained characteristic will not fully reflect the accuracy toward a real-world patient population because some tumors were not included in the analysis. On the other hand, in our study population, the ratios of prevalence of different tumor types are not like those in the real world, which means that the contribution of these tumor types to classifier error will still not reflect reality.

Therefore, we believe it would not be correct to recalculate the diagnostic characteristics of subtyping by including the tumor subtypes that the classifier cannot by definition identify correctly. Nonetheless, into Table 3, we added a column that shows the diagnostic characteristics of cancer detection in the study population that includes ATC and NIFTP. Changes to Figure 10 have also been made.

Example: among the 70 PTC analyzed (flow through this study, figure 8), 59 were real PTC at the final diagnosis. The specificity is lower than 100%.

This is true if ATC and NIFTP are included in the calculations.

A revision of the statistical analysis is mandatory.

Reviewer 5 Report

In this paper, Titov et al. aimed to include the identification of parathyroid cells in a previously developed diagnostic molecular classifier for thyroid tumors and to evaluated the accuracy of the test for typing different tumors in 278 Bethesda II-VI samples. The panel demonstrated a high accuracy in identifying parathyroid adenomas, Hürthle cell carcinoma, PTCs and MTCs, but not malignant follicular tumors (PPV 43.5%).

The authors have previously evaluated the diagnostic performance of the classifier on 122 indeterminate nodules (Titov et al, 2020). The originality of the present work is the implementation of the classifier with GCM2 expression as a marker of parathyroid adenomas. On the other hand, the clinical utility of identifying the type of thyroid tumors in cytological samples, including all Bethesda categories, is not quite as clear. In particular, the advantage of this classifier compared to traditional calcitonin screening for MTCs is not discussed.

Furthermore, the authors should better explain why they consider HCC a continuum of FTC, since the mtDNA/nDNA pattern is differnt in the two subtypes (line 319-322 and Figure 7).

Minor:

Line 109: Add reference.

Decision tree (figure 2) should be moved after paragraph 2.5.

Author Response

Review Report 5

In this paper, Titov et al. aimed to include the identification of parathyroid cells in a previously developed diagnostic molecular classifier for thyroid tumors and to evaluated the accuracy of the test for typing different tumors in 278 Bethesda II-VI samples. The panel demonstrated a high accuracy in identifying parathyroid adenomas, Hürthle cell carcinoma, PTCs and MTCs, but not malignant follicular tumors (PPV 43.5%).

The authors have previously evaluated the diagnostic performance of the classifier on 122 indeterminate nodules (Titov et al, 2020). The originality of the present work is the implementation of the classifier with GCM2 expression as a marker of parathyroid adenomas. On the other hand, the clinical utility of identifying the type of thyroid tumors in cytological samples, including all Bethesda categories, is not quite as clear. In particular, the advantage of this classifier compared to traditional calcitonin screening for MTCs is not discussed.

The main purpose of the molecular classifier is to detect cancer among Bethesda III-IV specimens, whereas determining the type of tumor is a supplementary aim that did not require inclusion of a large number of additional markers. Because there was an opportunity for subtyping, we wanted to test its accuracy, even if this does not apply to all the types of tumors that are worth identifying. In this study, we analyzed a sample of clinical specimens from all Bethesda categories simply because among specimens of Bethesda III-IV categories, it is difficult to find sufficient numbers of all the tumor subtypes under study.

In fact, at the moment, we are simply assessing the feasibility of molecular typing of tumors by means of cytological smears. Therefore, for now, we are determining the accuracy of such subtyping and the problems that arise, without addressing the clinical significance of this approach.

Furthermore, the authors should better explain why they consider HCC a continuum of FTC, since the mtDNA/nDNA pattern is differnt in the two subtypes (line 319-322 and Figure 7).

It is for this reason that we drew this conclusion because, if simultaneously with HCC specimens, there were many FTC specimens with an increased mtDNA/nDNA ratio, then this would indicate their independent origin. We believe that there are no (except one) outliers in the FTC group because they are all in the HCC group. On the other hand, these two groups differ in the expression of miRNA-221: this is described in the Discussion section (lines 331-333).

Minor:

Line 109: Add reference. The reference has been added.

Decision tree (figure 2) should be moved after paragraph 2.5.

The figure has been moved to paragraph 2.6.

Reviewer 6 Report

I think this work is very interesting and innovative because it aims to provide additional diagnostic tools in the field of undetermined thyroid cytologies. One of the most important challenges for endocrinologists in the last years is trying to optimize the surgical indication especially in nodules cytologically indeterminate.

Below I expose my considerations and questions for the authors:

  • Even though parathyroid tumors are rarely biopsied, because other diagnostic tool as scintigraphy are used for detecting parathyroid lesions, the discover of a molecular marker to distinguish between thyroid and parathyroid cells can be useful in selected rare cases of intrathyroid parathyroid glands or discrepancy between laboratory tests and parathyroid scintigraphy
  • Positive and negative predictive value of the study method is very high for papillary thyroid carcinoma, Hurtle cell carcinoma and medullary thyroid carcinoma.
  • This method, unfortunately, identify with lower accuracy the malignant follicular tumors, which already represent the most difficult malignant lesions to diagnose with cytological examination. (Indeterminate nodules Bethesda III e IV) . In this regard, why did you not include the Bethedsa III cytology nodules in the study?
  • What is the added value of the classifier in Bethesda IV and V nodules, already candidates for surgery?
  • I suggest to clarify the abbreviations in Figure 1.
  • Molecular analysis proposed by the authors is definitely an extra cost to the routine tests. It would be interesting to know from the authors if a cost analysis was also done to evaluate the cost / benefit of using this procedure.

Author Response

Review Report 6

I think this work is very interesting and innovative because it aims to provide additional diagnostic tools in the field of undetermined thyroid cytologies. One of the most important challenges for endocrinologists in the last years is trying to optimize the surgical indication especially in nodules cytologically indeterminate.

Below I expose my considerations and questions for the authors:

  • Even though parathyroid tumors are rarely biopsied, because other diagnostic tool as scintigraphy are used for detecting parathyroid lesions, the discover of a molecular marker to distinguish between thyroid and parathyroid cells can be useful in selected rare cases of intrathyroid parathyroid glands or discrepancy between laboratory tests and parathyroid scintigraphy
  • Positive and negative predictive value of the study method is very high for papillary thyroid carcinoma, Hurtle cell carcinoma and medullary thyroid carcinoma.
  • This method, unfortunately, identify with lower accuracy the malignant follicular tumors, which already represent the most difficult malignant lesions to diagnose with cytological examination. (Indeterminate nodules Bethesda III e IV). In this regard, why did you not include the Bethedsa III cytology nodules in the study? The reason is that in our work, we compared molecular diagnoses with histological ones, and we did not take cytological diagnoses into account. In addition, it would not be correct to say that the worst diagnostic characteristics were seen with Bethesda III and IV specimens; actually, the diagnostic characteristics were the worst for FTCs. As for not including Bethesda III in the analysis, this is because the morphologists who analyzed the specimens prefer to avoid this category.
  • What is the added value of the classifier in Bethesda IV and V nodules, already candidates for surgery? Indeed, the main purpose of the molecular classifier is to detect cancer among Bethesda III-IV specimens, whereas determining the tumor type is a supplementary aim that did not require inclusion of a large number of additional markers. Because there was an opportunity for subtyping, we wanted to test its accuracy. Among specimens of Bethesda III-IV categories, it is difficult to find sufficient numbers of all the tumor subtypes that we needed for this study; therefore, in this analysis, we included clinical specimens from all Bethesda categories.
  • I suggest to clarify the abbreviations in Figure 1. The definitions of the abbreviations were added into the Figure 1 legend: PTA, parathyroid adenoma; ATC, anaplastic thyroid carcinoma; MTC, medullary thyroid carcinoma; PTC, papillary thyroid carcinoma; NIFTP, noninvasive follicular thyroid neoplasm with papillarylike nuclear features; HCC, Hürthle cell carcinoma; FTC, follicular thyroid carcinoma; FTA, follicular thyroid adenoma.
  • Molecular analysis proposed by the authors is definitely an extra cost to the routine tests. It would be interesting to know from the authors if a cost analysis was also done to evaluate the cost / benefit of using this procedure. We have not yet investigated the economic issues of this classification method. Of course, such a method will impose additional costs on the patient. But in the case of an indeterminate cytological diagnosis, a new biopsy will be needed anyway, which also costs money, or an operation, which is even more expensive.

Round 2

Reviewer 4 Report

The authors answered all questions asked.

Reviewer 5 Report

It is still not clear why HCCs were considered a special subtype of FTC (line 324), rather than a distint group. Indeed, in their response, the autors stated that the increased mtDNA/nDNA ratio in HCCs would indicate their independent origin.